# Clustering of 27,525,663 Death Records from the United States Based on Health Conditions Associated with Death: An Example of Big Health Data Exploration

**DOI:** 10.3390/jcm8070922

**Published:** 2019-06-27

**Authors:** Daisy J.A. Janssen, Simon Rechberger, Emiel F.M. Wouters, Jos M.G.A. Schols, Miriam J. Johnson, David C. Currow, J. Randall Curtis, Martijn A. Spruit

**Affiliations:** 1Department of Research & Education, CIRO, Centre of expertise for chronic organ failure, 6085NM Horn, The Netherlands; 2Centre of Expertise for Palliative Care, Maastricht University Medical Centre (MUMC+), 6229HX Maastricht, The Netherlands; 3Department of Health Services Research, Maastricht University, 6229GT Maastricht, The Netherlands; 4Viscovery Software GmbH, 1130 Vienna, Austria; 5Department of Respiratory Medicine, Maastricht University Medical Centre (MUMC+), 6229HX Maastricht, The Netherlands; 6Department of Family Medicine, Maastricht University, 6229HA Maastricht, The Netherlands; 7Wolfson Palliative Care Research Centre, Hull and York Medical School, University of Hull, Hull HU6 7RX, UK; 8IMPACCT, Faculty of Health, University of Technology Sydney, Ultimo, NSW2007 New South Wales, Australia; 9Cambia Palliative Care Center of Excellence, Harborview Medical Center, University of Washington, Seattle, WA 98104, USA; 10NUTRIM School of Nutrition and Translational Research in Metabolism, 6229ER Maastricht, The Netherlands; 11REVAL-Rehabilitation Research Center, BIOMED-Biomedical Research Institute, Faculty of Rehabilitation Sciences, Hasselt University, BE3590 Diepenbeek, Belgium

**Keywords:** mortality, death, death certificates, palliative care, delivery of health care, multi-morbidity, ageing

## Abstract

Background: Insight into health conditions associated with death can inform healthcare policy. We aimed to cluster 27,525,663 deceased people based on the health conditions associated with death to study the associations between the health condition clusters, demographics, the recorded underlying cause and place of death. Methods: Data from all deaths in the United States registered between 2006 and 2016 from the National Vital Statistics System of the National Center for Health Statistics were analyzed. A self-organizing map (SOM) was used to create an ordered representation of the mortality data. Results: 16 clusters based on the health conditions associated with death were found showing significant differences in socio-demographics, place, and cause of death. Most people died at old age (73.1 (18.0) years) and had multiple health conditions. Chronic ischemic heart disease was the main cause of death. Most people died in the hospital or at home. Conclusions: The prevalence of multiple health conditions at death requires a shift from disease-oriented towards person-centred palliative care at the end of life, including timely advance care planning. Understanding differences in population-based patterns and clusters of end-of-life experiences is an important step toward developing a strategy for implementing population-based palliative care.

## 1. Introduction

The National Vital Statistics System of the National Center for Health Statistics uses death certificates in the United States (US) to register daily how people die in the US (since 1959). This allows clinicians and policy makers to better understand the death rates over time, but also permits study of associations between cause of death, place of death, age, gender, ethnic background, and known health conditions associated with death. Each death record consists of 117 attributes, most of them categorical with up to 7500 different categories per attribute. Although the data set from 2006 to 2016 consists of only 6 GB of data, the number of categorical attributes makes analyses by traditional processing application software challenging.

Self-organizing maps (SOMs) have been used to create an ordered representation of multi-dimensional data, to simplify complexity and to reveal meaningful associations. This method has been applied in various fields, like biology [1,2], biotechnology [3], sport sciences [4], social sciences [5], health sciences [6,7], business intelligence [8], text mining [9], and image recognition [10], to analyze big data and to visualize novel insights.

We aimed to cluster 27,525,663 deceased people based on entity codes (conditions that are able to be coded within the ICD10 coding scheme, further referred to as health conditions) that were associated with death using the data-mining tool Viscovery SOMine. This approach permitted study of the associations between health conditions clusters, demographics, the underlying cause of death, and the place of death. This can provide important knowledge to inform healthcare, including palliative care policy in the US. Indeed, the need for palliative care services will increase because of the ageing population and the increase in non-communicable diseases [11]. In 2015, life expectancy at birth was 78.7 years in the US. [12] Recent analyses of the Global Burden of Disease Study highlighted how improvements in early mortality for several conditions have resulted in older populations living with complex diseases. [13,14] This poses challenges for healthcare systems and requires integration of palliative care into existing health care services and within the continuum of care for persons with chronic, progressive life-limiting illnesses [15].

## 2. Materials and Methods

### 2.1. Data Source

For this study we analyzed the Multiple Cause-of-Death Mortality Data from 2006 to 2016 of the National Vital Statistics System of the National Center for Health Statistics, which provides data obtained from death certificates in the US (National Center for Health Statistics (2009–2017). Data File Documentations, Multiple Cause-of-Death, 2006–2016 (machine readable data file and documentation, CD-ROM Series 20, No. 2L–2V), National Center for Health Statistics, Hyattsville, Maryland). Data records without data in the health conditions (492,378 records (1.8% of all 28,018,041 records)) were omitted, resulting in a total number of 27,525,663 usable records. See additional file for details and Appendix A for accessible attributes.

### 2.2. Definitions

Health conditions associated with death and underlying primary cause of death were coded using the International Classification of Diseases (ICD)-10 coding scheme (1992 Revision, Volume 1 for years 2006–2009 and 2004 Revision, Volume 1 for years 2010–2016). These codes were aggregated into 346 disease categories (Appendix A). A multi-morbidity indicator was defined: one if the number of different health conditions is two or more; zero otherwise. Manner of death was categorized as natural death, homicide, suicide, or accident. See Appendix A for coding of educational level (Appendix A), age, race, resident status, and place of death.

### 2.3. Statistics

Statistical analyses were performed using Viscovery SOMine 7.1 by Viscovery Software GmbH (www.viscovery.net; Vienna, Austria). A SOM was used to create an ordered representation of the mortality data. SOMs are a form of unsupervised artificial neural network and can be viewed as a non-parametric regression technique that converts multi-dimensional data spaces into lower dimensional abstractions. A SOM consists of units on a grid (nodes), where each node represents a group of very similar data records. Thereby, the SOM generates a nonlinear representation of the data distribution, which can be used to identify homogenous data groups and perform statistical analyses on them [16]. Deceased persons have been ordered by the 346 different health conditions reported at the time of death (Appendix A). Based on the created SOM model, clusters have been generated using the SOM-Ward Cluster algorithm of Viscovery, a hybrid algorithm that superimposes the classical hierarchical cluster method of Ward on the SOM topology. Attributes are visualized by depicting the average value for each node on a fitting color scale. For every cluster, all categorical variables are given in absolute numbers and percentages. Continuous variables are presented with mean and standard deviation. Cluster means are tested with a two-sided t-test against the global mean. Given the size of the dataset, a priori, a *p*-value <0.01 was considered statistically significant.

## 3. Results

### 3.1. Total Sample

In total, 27,525,663 death records were available (49.8% women; 85.5% white; mean age at the time of death of 73.1 (18.0) years) (Table 1). At death, most deceased were married (38.0%) or widowed (35.4%). Most reported sites of death were hospital inpatient (32.6%), at home (27.8%), or nursing home (20.7%). Most persons died from a natural cause (90.9%). Atherosclerotic heart disease (9.3%) was the main cause of death (Appendix A). At death, a mean of 3.0 (1.8) health conditions were reported, resulting in multi-morbidity in 78.4% of subjects with one or more health conditions. Atherosclerotic heart disease was the most reported health condition at the time of death (code present in any entity field, 16.4%), followed by cardiac arrest (13.5%), hypertension (12.9%), heart failure (12.1%), and chronic obstructive pulmonary disease (COPD; 10.7%) (Appendix A).

### 3.2. Clusters Based on the Health Conditions at the Time of Death

Sixteen clusters were identified (Figure 1), which all showed significant differences in health conditions at death (Appendix A), socio-demographics (Figure 2), place of death (Figure 3), causes of death (Appendix A), and manner of death (Figure 4) compared to the whole sample. The five largest clusters are described below. The other 11 clusters are described in Appendix A. All details about the clusters can be found in Appendix A.

#### 3.2.1. The ‘Other Cardiovascular Disease’ Cluster (*n* = 4,850,234)

The top-five most prevalent health conditions associated with death in this cluster were (1) heart failure (34.1%); (2) atherosclerotic heart disease (22.5%); (3) cardiac arrest (21.8%); (4) hypertension (16.6%); and (5) diabetes mellitus (14.8%). The top-five most prevalent underlying primary causes of death were: (1) atherosclerotic heart disease (15.8%); (2) heart failure (10.0%); (3) hypertensive heart disease (6.2%); (4) diabetes mellitus (5.2%); and (5) myocardial infarction (4.8%). Compared to the whole population, the deceased in this cluster had a significantly higher mean age at the time of death and were more often men, black, widowed, and local residents. 43.5% died in the hospital and 28.6% died at home. The deceased in this cluster had a higher prevalence of multi-morbidity than the whole population.

#### 3.2.2. The ‘Other Cancer’ Cluster (*n* = 3,316,586)

The top-five most prevalent health conditions associated with death in this cluster were (1) other cancer (15.8%); (2) breast cancer (13.6%); (3) prostate cancer (10.9%); (4) lung cancer (9.9%); and (5) cardiac arrest (9.6%). The top-five most prevalent underlying primary causes of death were: (1) breast cancer (12.1); (2) other cancer (9.6%); (3) prostate cancer (8.4%); (4) lung cancer (6.3%); and non-Hodgkin lymphoma (6.2%). Compared to the whole population, the deceased in this cluster had a significantly lower mean age at the time of death; were more often men, white, married, local resident; and were more likely to have tertiary education level. They more often died at home or in a hospice facility, and died due to a natural cause. Moreover, the deceased had a lower prevalence of multimorbidity than the whole population.

#### 3.2.3. The ‘Urological/Gastrointestinal Disease/Other Infection’ Cluster (*n* = 3,125,752)

The top-five most prevalent health conditions associated with death in this cluster were (1) other sepsis (29.7%); (2) unspecified kidney failure (16.1%); (3) cardiac arrest (14.3%); (4) respiratory failure (13.9%); and (5) diabetes mellitus (11.7%). The top-five most prevalent underlying primary causes of death were: (1) other sepsis (8.0%); (2) dementia (4.6%); (3) atherosclerotic heart disease (4.6%); (4) diabetes mellitus (4.2%); and (5) unspecified kidney failure (3.9%). Compared to the whole population, the deceased in this cluster had a significantly higher mean age at the time of death; were more often women, black or American Indian, widowed or never married, non-local US resident, more often had only primary education, and were more likely to die in the hospital (inpatient, 58.4%). Moreover, the deceased had a higher prevalence of multi-morbidity than the whole sample.

#### 3.2.4. The ‘Respiratory Disease’ Cluster (*n* = 2,561,110)

The top-five most prevalent health conditions associated with death in this cluster were (1) chronic obstructive pulmonary disease (46.8%); (2) pneumonia (35.3%); (3) respiratory failure (28.2%); (4) nicotine dependence (23.7%); and (5) cardiac arrest (12.2%). Moreover, the top-five most prevalent underlying primary causes of death were (1) chronic obstructive pulmonary disease (37.7%); (2) pneumonia (16.7%); (3) other interstitial pulmonary disorder (6.2%); (4) atherosclerotic heart disease (5.3%); and (5) lung cancer (2.5%). Compared to the whole population, the deceased in this cluster had a significantly higher mean age at the time of death; were more often women; were more often white; were less likely to have tertiary education level; were more likely to be a non-local US resident; more often died in the hospital (inpatient, 46.6%); and more often died due to a natural cause. Moreover, the deceased had more health conditions associated with death, and a higher prevalence of multimorbidity.

#### 3.2.5. The ‘Other Geriatric Disease’ Cluster (*n* = 1,833,980)

The top-five most prevalent health conditions associated with death in this cluster were (1) dementia (65.6%); (2) hypertension (19.5%); (3) pneumonitis (17.9%); (4) atherosclerotic heart disease (14.5%); and (5) cardiac arrest (12.3%). Moreover, the top-five most prevalent underlying primary causes of death were: (1) dementia (47.7%); (2) atherosclerotic heart disease (7.8%); (3) pneumonitis (7.0%); (4) unspecified stroke (6.0%); and (5) chronic obstructive pulmonary disease (4.1%). Compared to the whole population, the deceased in this cluster had a significantly higher mean age at the time of death; had a higher proportion of women; were more often white; more often had received only primary education; were more often widowed; were more likely to be a local resident; more often died in a nursing home (50.4%); and more often died due to a natural cause. Moreover, the deceased had more health conditions associated with death; and a higher prevalence of multimorbidity.

## 4. Discussion

Significant differences in socio-demographics, place of death, and the primary underlying cause of death were found after clustering 27,525,663 death records based on the health condition(s) associated with death. Generally, most people died at old age and had multiple health conditions associated with death. The lowest proportion of subjects died in a hospice facility, and most people died in the hospital or at home.

### 4.1. Big Health Data Exploration

This analysis showed that death record data from more than 27 million deceased subjects can be clustered based on one or more of the possible 346 health conditions associated with death using the data-mining tool Viscovery SOMine. Clustering data sets of this size and dimensionality is not an easy task for many conventional cluster algorithms due to computational load. The Viscovery SOMine suite allows to represent large data sets on a SOM calculated from a random sample and uses this low-dimensional representation to conduct cluster analysis and group profiling on the entire data. The SOM allows a visualization of the findings in an intuitive way. For example, Figure 3 clearly shows that the highest proportion of subjects dying while being inpatients in the hospital are in cluster ‘Urological/gastrointestinal disease/other infection’. ‘Other cardiovascular disease’ was the largest cluster and chronic ischemic heart disease was the main cause of death. These findings are in line with the predictions of the Global Burden of Disease Study (GBDS) 2017 [14].

### 4.2. Clinical Relevance of Current Findings

#### 4.2.1. Multiple Health Conditions Associated with Death

Chronic life-limiting diseases are highly prevalent among both causes of death as well as among the health conditions associated with death. The deceased had a mean of three health conditions associated with death, and about three quarters had ≥2 health conditions associated with death. This corroborates the GBDS 2017 findings that improvements in premature mortality for select conditions have led to older populations with complex diseases [13]. Indeed, the presence of multiple health conditions is associated with poor quality of life, reduced life expectancy, treatment burden, and polypharmacy [17]. So, there is a great need for interdisciplinary guidelines for people with multimorbidity. While healthcare delivery is often single-disease-oriented [18], palliative healthcare professionals cannot focus on addressing a single health condition, and use a person-centred approach to address palliative care needs in patients with multiple health conditions and limited life expectancy. Moreover, interdisciplinary collaboration and communication is paramount [18]. For example, a recent systematic review exploring the evidence for palliative care for patients with heart failure found that access to specialist palliative care, in addition to cardiac care, appears beneficial, although there are no multi-site, generalizable studies yet. Emerging models where most palliative care is delivered as part of disease-oriented care, with referral to specialist services if needed are encouraging but need confirmation in clinical trials [19].

#### 4.2.2. Dying at Old Age

In line with global mortality data [14], the current analysis showed that most people die at old age. Indeed, 73.2% of the deceased persons (*n* = 20,132,389) were aged 65 years or older; and 30.7% (*n* = 8,451,749) were aged 85 years or older at death. Healthcare organizations and healthcare professionals need to be prepared to offer palliative care to this growing number of dying elderly. Indeed, projections from mortality statistics of England and Wales from 2006 to 2014 predict an increase in the proportion of people dying above 85 years from 38.8% in 2014 to 53.2% in 2040 [20].

The mean age at death in the cluster ‘Geriatric disease’ was 85 years and dementia was the most prevalent cause of death within this cluster. Previous studies have shown that dementia is a life-limiting illness and persons with dementia are in need of palliative care [21]. However, providing palliative care to patients with dementia can be challenging and may require involvement of palliative care specialists [22,23]. The presence of multiple health conditions may even be an additional complicating factor [24]. Of the 2,287,046 deceased with dementia as a reported health condition in the current sample, 1,992,252 (87.1%) also had one or more other health conditions.

#### 4.2.3. Dying at Young Age

In the current sample, 5.4% of the deceased (*n* = 1,497,487) had an age of 40 years or younger. The clusters ‘Open wound/suffocation’, ‘Poisoning’, and ‘Other physical harm’ had the highest proportion of younger people: 52.0%, 41.8%, and 27.8%, respectively. This is most probably related to the manner of death (e.g., accident, suicide, or homicide) associated with the death causes constituting these clusters. However, younger people were also represented in the other clusters (in total 710,773 deceased). This is important to realize, as palliative care needs of younger people with life-limiting illnesses may differ from palliative care needs from the elderly [25]. In the current sample, 322,847 death records (1.2%) were from children aged below 18 years at the time of death. This is a challenging subgroup for palliative care, as the evidence base for pediatric palliative care is less advanced and compassion fatigue and burn out occur frequently in pediatric palliative care providers [26].

#### 4.2.4. Other Socio-Demographic Differences between Clusters

Other socio-demographic differences between clusters need to be considered to prevent disparities in palliative care uptake as well as to address specific palliative care needs.

First, gender differences were found in all clusters, except chronic kidney disease. For example, men are more likely than women to die from an open wound, suffocation, or other physical harm, while women are more likely to die from neurodegenerative or other geriatric diseases. Gender differences are important to consider to prevent disparities in palliative care uptake. Indeed, a recent study showed significant differences between male and female patients with advanced cancer in preference for palliative care. Women were more likely to prefer palliative care than men [27].

Second, differences in educational level were found between all clusters. Figure 2 clearly shows that people dying from open wounds, suffocation, or poisoning were the least likely to have followed only primary education. Patients dying from other cancer were the most likely to have followed tertiary education, while patients dying from respiratory diseases were the least likely to have followed tertiary education. Individuals with lower educational attainment are at higher risk for health illiteracy, and may have skills to find, understand, and apply information about healthcare. Health illiteracy, therefore, negatively affects the ability to participate in shared-decision making as well as advance care planning. Therefore, clinicians need other strategies and/or tools to support communication and decision-making in palliative care in people with health illiteracy [28].

Third, racial differences should be considered. In fact, Native Americans were overrepresented in the clusters liver disease and other physical harm compared to the other clusters. People who are Asian or Pacific Islander were overrepresented in the clusters stroke and gastro-intestinal cancer compared to other clusters, while black people were overrepresented in the cluster open wound/suffocation and underrepresented in the clusters neurodegenerative disease and other geriatric diseases. Race seems to influence the provision of palliative care as well as healthcare intensity at the end of life, with racial/ethnic minorities receiving higher intensity care, such as ICU admission or life-sustaining treatments at the end of life [29]. Among patients with cancer, being black, Hispanic, or Asian is associated with dying in the hospital [30]. A US study showed that black children dying from cancer more often received CPR than white children. Hispanic children less frequently received cancer therapy 28 days prior to death than non-Hispanic children [31]. Moreover, palliative care clinicians seem more reluctant to discuss prognosis with black or Latino patients with cancer than with white patients [32]. Several explanations for these racial differences have been hypothesized, including differences in preferences regarding life-sustaining treatments, but also a lower likelihood of end-of-life preferences being discussed and honored [31].

Fourth, the present study also showed differences in marital status between clusters, which may need to be considered in organizing palliative care. For example, subjects in clusters ‘Gastrointestinal cancer’ (49%), ‘Lung cancer’ (49%), and ‘Other cancer’ (50%) were more often married at time of death, while subjects in the clusters ‘Geriatric disease’ (58%) and ‘Neurodegenerative disease’ (50%) were more often widowed. A study performed in the United Kingdom showed that older people live years (men: 2.4 years; women 3.0 years) with substantial care needs [33]. Partners of persons with disabilities often contribute significantly in providing care, but the burden for family caregivers can be considerable [34]. Persons with dementia are less likely to have an informal caregiver available [35]. On the other hand, family caregivers of persons with dementia and disabilities experienced the highest burden [35]. Palliative care interventions have been shown to be effective in reducing burden for family caregivers [36]. In an English household survey, family caregivers at the end of life were more likely to be willing to care again under the same circumstances if the decedent had had access to palliative care services [37].

#### 4.2.5. Place of Death

Dying at the place of preference is seen as one of the key principles of a good death [38]. Home is often seen as the most preferable place to die [39]. Indeed, a study performed in the US showed that patients who received end-of-life care consistent with their preferences were more likely to die at home than patients who received care inconsistent with their wishes [40]. Despite this, only 28% of the deceased subjects in the current study died at home. Subjects in the clusters ‘Gastrointestinal cancer’ (45%) and ‘Lung cancer’ (44%) were more likely to die at home than the whole group, while persons in the clusters ‘Urological/gastrointestinal disease/other infection’ (58%), ‘Respiratory disease’ (47%), ‘Liver disease’ (45%), and ‘Stroke’ (44%) were more likely to die as inpatients in the hospital. This might reflect the predictability of death and the attention for timely initiation of palliative care and advance care planning. Indeed, people with non-malignant disease have less access to palliative care than patients with malignant diseases, and in the English household survey, access to palliative care reduced the proportion of hospital deaths and increased that of those dying at home. [41] In the past, cancer was seen as a disease with a predictable disease trajectory, resulting in timely palliative care and advance care planning [42]. This could explain the higher number of home deaths for patients with cancer in comparison with chronic non-malignant diseases [43]. However, due to new developments in cancer treatment, several forms of cancer may become a chronic disease with a more complex disease trajectory characterized by uncertainty [44]. Then again, the current data emphasize the need for timely advance care planning in patients with chronic life-limiting illnesses, such as chronic respiratory diseases and liver diseases, to assure that end-of-life preferences are met [45].

### 4.3. Methodological Considerations

The strength of the current study does not lie in the novelty of any disease specific findings. The novelty of our manuscript lies in our approach and the strong statistic confirmation of many more or less known facts, derived from findings of often smaller or specialized studies. Indeed, we included data of 27,525,663 death records that represent a complete population-based sample of death in the US. The strength of this Big Data approach is seeing the “grand picture”. This is also why we chose not to stratify the data with respect to gender, race, or any other attribute, since we wanted to keep the focus on the entire population, instead of specific subgroups.

The Viscovery software provided innovative data representation and visualization SOMs, clustering as well as statistical profiling of these death records. The use of SOMs in general and the Viscovery software in particular has several advantages for the analysis of this mortality data set. Since the data is large in quantity as well as complexity we wanted a high-performance method to accurately represent the data distribution, while reducing complexity to a handleable degree, which is one of the strengths of SOMs. In addition, this approach allows for innovative visualization to study interdependencies between diseases and other information. Furthermore, the map representation can be used to apply hierarchical clustering methods to this huge data set, which would be impractical with classical methods because of their high running time complexity of O (*n*^2^) or worse.

The resulting cluster model is the most complete picture we know about common comorbidity groups in US citizens deceased between 2006 and 2016 and their association with socio-demographic patterns and other “meta-data”, like place of death. However, the following limitations need to be considered. First, we rely on the information provided in death records of the National Center for Health Statistics and this information may not always be correct. For example, studies performed in New York showed over-reporting of deaths from coronary heart disease, which decreased after training [46,47]. Indeed, autopsies may show other causes of death and previously unsuspected findings [48]. Cardiac arrest was reported in a subgroup of patients while, obviously, cardiac arrest is present in all deaths. Moreover, cause of death as reported in death certificates might be even less accurate in very old persons [49]. Despite low agreement between hospital diagnoses and recorded causes of death, agreement at population level is reasonable [50]. However, errors may not only occur in causes of death, but also in data like place of death [51]. Second, sometimes data were missing, and not all data of interest were available. For example, data on religious background, and medication use were not available. Third, we only have information at time of death, but we do not have any information about the trajectory of health conditions before death and this information is also paramount for the development of optimal healthcare systems. Finally, we only included data from the US and therefore, the current findings cannot be extrapolated to all other parts of the world. Indeed, important international differences are present for causes of death, place of death, and organization of palliative care [43,52,53].

## 5. Conclusions

Clustering of more than 27 million death records based on the health conditions at the time of death resulted in 16 distinct clusters, with significant differences in socio-demographics, place of death, the cause of death, and, obviously, the health condition(s) at the time of death. Healthcare organizations and professionals need to be prepared to offer interdisciplinary palliative care to a growing number of people dying with multiple health conditions. Indeed, the high prevalence of multiple health conditions requires a shift from primarily disease-oriented care towards person-centred palliative care at the end of life, including timely advance care planning [19]. Understanding differences in population-based patterns and clusters of end-of-life experiences for patients and their families is an important step toward portend developing a strategy for implementing population-based palliative care.

## Figures and Tables

**Figure 1 jcm-08-00922-f001:**
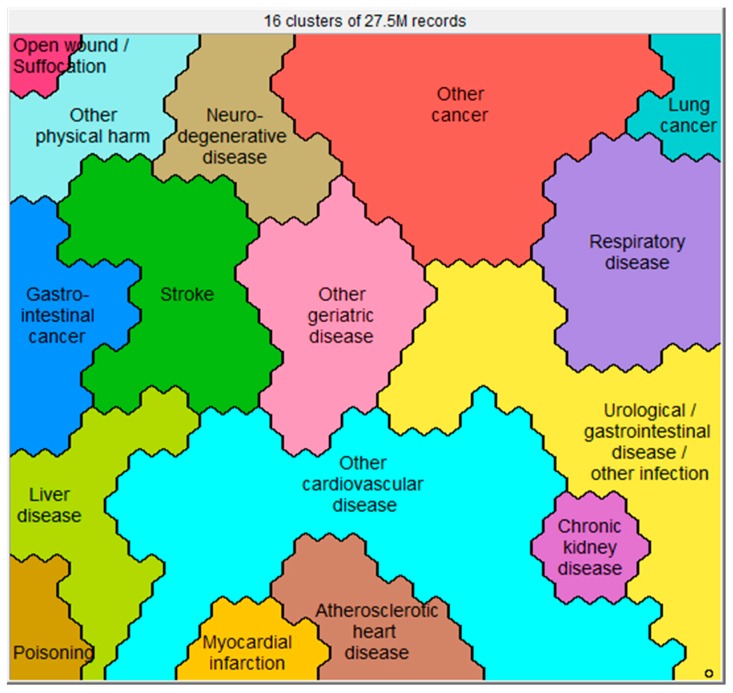
Sixteen clusters based on health conditions at death as reported in 27,525,663 death records. Subjects resembling each other in terms of these characteristics are close to each other on the map.

**Figure 2 jcm-08-00922-f002:**
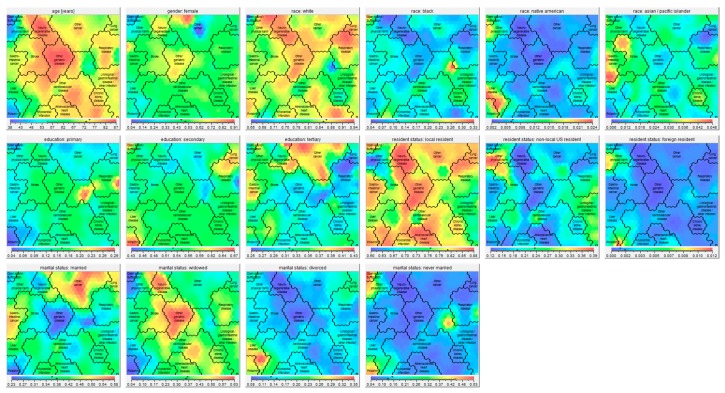
Self-organizing maps showing the socio-demographics (age, gender, race, education, resident status, and marital status) of the 16 clusters. Age: Red parts of the map consist of persons with a high average age at death, whereas blue parts consist of persons with a low average age at death. The exact average ages are given in the bar under the map picture. Gender, race, education, resident status, marital status: red parts of the map consist of high percentages of persons of the indicated gender, race, education, resident status, and marital status, whereas blue parts consist of low percentages of the indicated gender, race, education, resident status, and marital status. The exact percentages are given in the bar under the respective map picture.

**Figure 3 jcm-08-00922-f003:**
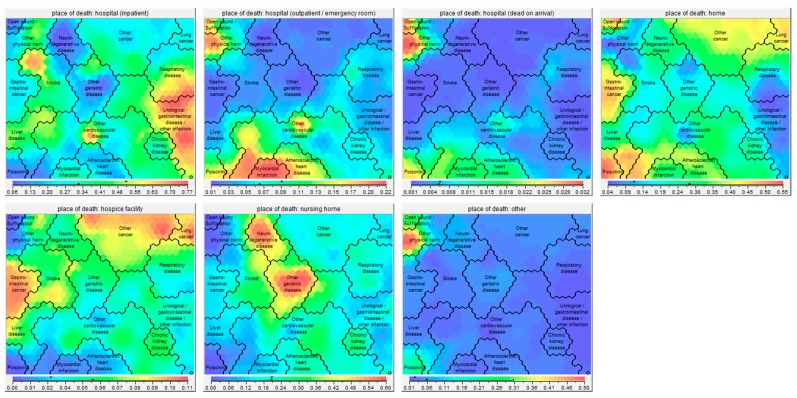
Self-organizing maps showing the place of death (hospital, home, hospice facility, nursing home, other) of the 16 clusters. Red parts of the map consist of high percentages of persons dying in the indicated place, whereas blue parts consist of low percentages of persons dying in the indicated place. The exact percentages are given in the bar under the respective map picture.

**Figure 4 jcm-08-00922-f004:**
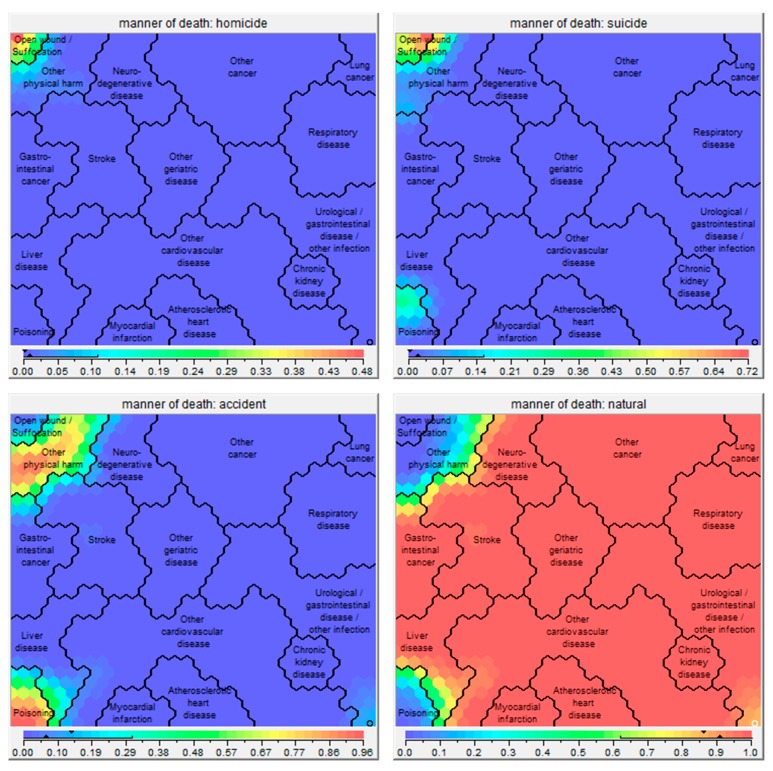
Self-organizing maps showing the manner of death (homicide, suicide, accident, natural death) of the 16 clusters. Red parts of the map consist of high percentages of persons dying in the indicated manner, whereas blue parts consist of low percentages of persons dying in the indicated manner. The exact percentages are given in the bar under the respective map picture.

**Table 1 jcm-08-00922-t001:** Characteristics as reported in 27,525,663 death records.

Person Characteristics	n (%) or Mean (SD)
Age at death (years)	73.12 (18.00)
Female	13,697,099 (49.76%)
**Race**	
White	23,533,801 (85.50%)
Black	3,213,637 (11.68%)
American Indian	176,329 (0.64%)
Asian/Pacific islander	601,896 (2.19%)
**Education Level**	
Primary	3,536,306 (13.29%)
Secondary	14,546,015 (54.65%)
Tertiary	8,532,106 (32.06%)
**Marital Status**	
Married	10,359,202 (38.02%)
Widowed	9,633,551 (35.36%)
Divorced	3,930,144 (14.43%)
Never married	3,320,267 (12.19%)
**Resident Status**	
Local Resident	22,265,873 (80.89%)
Non-Local US Resident	5,211,399 (18.93%)
Foreign Resident	48,391 (0.18%)
**Place of Death**	
Hospital (Inpatient)	8,929,942 (32.61%)
Hospital (Outpatient / Emergency room)	1,851,801 (6.76%)
Hospital (Dead on Arrival)	163,868 (0.60%)
Home	7,609,594 (27.79%)
Hospice Facility	1,450,220 (5.30%)
Nursing Home	5,658,607 (20.66%)
Other	1,722,917 (6.29%)
**Manner of Death**	
Accident	1,394,442 (6.26%)
Suicide	435,553 (1.96%)
Homicide	201,661 (0.91%)
Natural	20,243,277 (90.88%)
Different Health Conditions associated with Death ^a^	3.04 (1.83)
Multimorbid ^b^	21,579,345 (78.40%)

^a^ number of codes on death certificate; ^b^ 2 or more different health conditions Source: National Center for Health Statistics (2006–2016).

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
