# Peer review of "Clustering of 27,525,663 Death Records from the United States Based on Health Conditions Associated with Death: An Example of Big Health Data Exploration"

_jcm, 2019, doi:10.3390/jcm8070922_

Reviewer 1 Report

Yes, the manuscript has been significantly improved.

Author Response

C1. The manuscript has been significantly improved

R1. Thank you.

Reviewer 2 Report

An interesting and large scale study. Overall presents clear indication of how/where people are dying in the USA in the given time period. It is pertinent that this presents findings whereby the most prolific diseases are cardiac and respiratory related and are the areas where palliative care resources and certainly lacking on a global scale. 

The graphics presented by SOM are excellent and easy to read. 

Although in the discussion there is some mention of the need for palliative care services to support those with CHF/COPD etc., there are no suggestions as to what solutions could be presented to rectify this problem. While the statistical data is useful to know, I do ask 'so what?' in terms of what does this information mean in real life situations of healthcare practices in the USA and overseas? How do we reach such high numbers of individuals with these conditions to support them? While you may not have the answers, it would be helpful to indicate how your study data can be used to move forward as opposed to just saying we need more palliative care interventions. 

Author Response

C1. An interesting and large scale study. Overall presents clear indication of how/where people are dying in the USA in the given time period. It is pertinent that this presents findings whereby the most prolific diseases are cardiac and respiratory related and are the areas where palliative care resources and certainly lacking on a global scale. 

The graphics presented by SOM are excellent and easy to read. 

R1. Thank you.

C2. Although in the discussion there is some mention of the need for palliative care services to support those with CHF/COPD etc., there are no suggestions as to what solutions could be presented to rectify this problem. While the statistical data is useful to know, I do ask 'so what?' in terms of what does this information mean in real life situations of healthcare practices in the USA and overseas? How do we reach such high numbers of individuals with these conditions to support them? While you may not have the answers, it would be helpful to indicate how your study data can be used to move forward as opposed to just saying we need more palliative care interventions. 

R2. To address this comment we have added to the discussion (4.2.1): “While healthcare delivery is often single-disease-oriented[18], palliative healthcare professionals cannot focus on addressing a single health condition, and use a person-centred approach to address palliative care needs in patients with multiple health conditions and limited life expectancy. Moreover, interdisciplinary collaboration and communication is paramount[18]. For example, a recent systematic review exploring the evidence for palliative care for patients with heart failure found that access to specialist palliative care, in addition to cardiac care, appears beneficial although there are no multi-site, generalizable studies yet. Emerging models where most palliative care is delivered as part of disease-oriented care, with referral to specialist services if needed are encouraging but need confirmation in clinical trials.[19]”

This manuscript is a resubmission of an earlier submission. The following is a list of the peer review reports and author responses from that submission.

Round  1

Reviewer 1 Report

This is an interesting exploration of mortality data, but there are significant limitations that limit its usefulness and interpretability.

Throughout:

The term “health conditions” should not be used. The codes on the file are either underlying cause of death (UCOD) or entity codes. Entity codes specifically refer to conditions that are able to be coded.

Intro:

First paragraph, the authors should cite NCHS and also evaluations of US mortality conducted to date

Data Sources:

Do the authors use the Multiple Cause of Death file? It is unclear from the language and not cited.

Definitions:

How were the 346 code aggregations formed?

Does the “mulit-morbidity” indicator mean that 2+ entity codes were present? Or does that mean the UCOD plus 1 entity?

Statistics:

Table A4 is very difficult to understand. Some of the rows (are these UCOD?) contain values that are not appropriate as UCOD, such as the S and T codes

Why are the p-values percents?

What does the red shading mean?

Does “health condition at time of death” mean the code was present in any entity field?

Rather than exhaustively review the clusters, the authors should highlight  what novel findings there are. A3 and A4 could still be included to show the process.

The authors need to discuss in much greater detail the limitations of these data and using them in these types of clusters. There are many citations regarding the incompleteness and inaccuracy of death certificate data, especially when state or county are considered.

I suggest the authors selectively present results as a way in demonstrating the usefulness of the Viscovery SOMine package and include only those findings that are most interesting or novel, rather than those that have been extremely well-documented in the literature.

Author Response

Reviewer 1

This is an interesting exploration of mortality data, but there are significant limitations that limit its usefulness and interpretability.

C1. Throughout:

The term “health conditions” should not be used. The codes on the file are either underlying cause of death (UCOD) or entity codes. Entity codes specifically refer to conditions that are able to be coded.

R1. We have used the entity codes. Since using the term “entity codes” throughout the manuscript might be difficult to understand for readers we opted to call them “health conditions”. We have now explained this more precisely in the introduction: “We aimed to cluster 27,525,663 deceased people based on entity codes (conditions that are able to be codedwithin the ICD10 coding scheme, further referred to as health conditions)” (line 61-62 of the revised manuscript with changes marked)

C2. Intro:

First paragraph, the authors should cite NCHS and also evaluations of US mortality conducted to date.

R2. We have included in the introduction: “In 2015, life expectancy at birth was 78.7 years in the US.[12]Recent analyses of the Global Burden of Disease Study highlighted how improvements in early mortality for several conditions have resulted in older populations living with complex diseases.[13,14]” (line 67-70 of the revised manuscript)

C3. Data Sources: Do the authors use the Multiple Cause of Death file? It is unclear from the language and not cited.

R3. Yes we used the multiple cause of death file. We have added the following to the method section: “For this study we analyzed the Multiple Cause-of-Death Mortality Data from 2006 to 2016 of the National Vital Statistics System of the National Center for Health Statistics from death certificate data in the US.(National Center for Health Statistics (2009-2017). Data File Documentations,Multiple Cause-of-Death , 2006-2016 (machine readable data file and documentation, CD-ROM Series 20, No. 2L-2V ), National Center for Health Statistics, Hyattsville, Maryland)” (line 76-81 of the revised manuscript)

We also included the following statement: Analyses, interpretations, or conclusions were from the authors and not to NCHS, which is responsible only for the initial data. (line 558-559 of the revised manuscript)

Further: we included in each table legend: Source: National Center for Health Statistics (2006-2016).

C4. Definitions:

How were the 346 code aggregations formed?

R4. We have explained how the 346 code aggregations were formed in the revised version of appendix A: “The aggregations were made from a data analytic point of view with additional consideration of medical connections and the structure of the ICD10 codes. First all codes in the entity variables of the data set where listed and their frequency of occurrence was considered. Relatively frequent codes (>0.05% prevalence in the data set) were considered important for the analysis. Less frequent codes where added to similar more frequent ones, whenever possible. When this was not possible, a more general group was considered for specific codes or codes where aggregated into leftover categories, such as "Other infection" or "Other metabolic disease".” (line 579-586 of the revised manuscript)

C5. Does the “mulit-morbidity” indicator mean that 2+ entity codes were present? Or does that mean the UCOD plus 1 entity?

R5. We used the number of entity-axis conditions indicator “eanum”. Multimorbidity means that eanum>1. This means that 2+ entity codes are present, where ucod coincides with one of these. 

In the main manuscript is mentioned: A multi-morbidity indicator was defined: one if the number of different health conditions is two or more; zero otherwise. (line 88-89 of the revised manuscript)

We have explained this in more detail in Appendix A: Finally, a multimorbidity indicator was defined, which is 1 if the number of different health conditions (eanum) is two or more (ucodand one or more additional entity codes), and 0 otherwise. (line 595-597 of the revised manuscript)

C6. Statistics:

Table A4 is very difficult to understand. Some of the rows (are these UCOD?) contain values that are not appropriate as UCOD, such as the S and T codes

Why are the p-values percents?

What does the red shading mean?

R6. Table A4 contains ucod = “Cause of Death” and entities = “Health conditions at Death”. Since codegroups for ucod are the same as for entities, S and T appear in both cases, but are 0 for ucod. The ICD10 coding instructions state that codes starting with V-Y which are associated to external causes (car accident, exposure to fire, etc.) are predominant over S and T for the underlying cause of death.  

We changed the p-values from percents to the absolute value. A red-colored cell in the columns that show the P-values means that the cluster-specific value is not significantly different from the value of the whole sample. (please see legend Table A4)

C7. Does “health condition at time of death” mean the code was present in any entity field?

R7. Yes, it does. We have explained this the first time we used “health condition at time of death”: “Atherosclerotic heart disease was the most reported health condition at the time of death (code present in any entity field, 16.4%)..”. (line 119-120 of the revised manuscript)

C8. Rather than exhaustively review the clusters, the authors should highlight  what novel findings there are. A3 and A4 could still be included to show the process.

R8. We understand the issue from the reviewer. The novelty of this manuscript lies not in the novel findings; rather the novelty of our approach lies in the strong statistic confirmation of many more or less known facts, derived from findings of usually smaller or specialized studies. We have acknowledged this in the discussion of the revised manuscript. Please see R10. 

In addition, in response to this comment we have kept the description of the five largest clusters in the main manuscript and have described the other 11 clusters in Appendix B. 

C9. The authors need to discuss in much greater detail the limitations of these data and using them in these types of clusters. There are many citations regarding the incompleteness and inaccuracy of death certificate data, especially when state or county are considered.

R9. We agree that the data have limitations. Therefore, we have discussed these more extensively in the methodological considerations section: “First, we rely on the information provided in death records of the National Center for Health Statistics and this information may not always be correct. For example, studies performed in New York showed over reporting of deaths from coronary heart disease, which decreased after training.[39,40]Indeed, autopsies may show other causes of death and previously unsuspected findings[41]. Cardiac arrest was reported in a subgroup of patients while obviously, cardiac arrest is present in all deaths. Moreover, cause of death as reported in death certificates might be even less accurate in very old persons.[42]Despite low agreement between hospital diagnoses and recorded causes of death, agreement at population level is reasonable.[43]However, errors may not only occur in causes of death, but also in data like place of death.[44]Second, sometimes data were missing, and not all data of interest were available. For example, data on religious background, and medication use were not available.” (line 512-522 of the revised manuscript)

Incompleteness/inaccuracy of state and county data is of limited concern, while, we only use that in form of the resident status attribute, which does not contribute to the ordering of the model. 

C10. I suggest the authors selectively present results as a way in demonstrating the usefulness of the Viscovery SOMine package and include only those findings that are most interesting or novel, rather than those that have been extremely well-documented in the literature.

R10. We have made changes to address this comment. Please see R8.

Further, we have added the following to the discussion to address this point: “The strength of the current study does not lie in the novelty of any disease specific findings. The novelty of our manuscript lies in our approach and the strong statistic confirmation of many more or less known facts, derived from findings of often smaller or specialized studies. Indeed, we included data of 27,525,663 death records that represents a complete population-based sample of death in the US. The strength of this Big Data approach is seeing the "grand picture". This is also why we chose not to stratify the data with respect to gender, race or any other attribute, since we wanted to keep the focus on the entire population, instead of specific subgroups.

The Viscovery software provided innovative data representation and visualization SOMs, clustering as well as statistical profiling of these death records. The use of SOMs in general and the Viscovery software in particular has several advantages for the analysis of this mortality data set. Since the data is large in quantity as well as complexity we wanted a high-performance method to accurately represent the data distribution, while reducing complexity to a handleable degree, which is one of the strengths of self-organizing maps. In addition, this approach allows for innovative visualization to study interdependencies between diseases and other information. Furthermore the map representation can be used to apply hierarchical clustering methods to this huge data set, which would be impractical with classical methods because of their high running time complexity of O(n2) or worse.

The resulting cluster model is the most complete picture we know about common comorbidity groups in US citizens deceased between 2006 and 2016 and their association to socio-demographic patterns and other "meta-data", like place of death. However, the following limitations need to be considered. […]” (line 486-512 of the revised manuscript)

Reviewer 2 Report

This is an interesting study to analyze a very large mortality dataset at the national level in a new way. I think the goal of study is significant and method used is scientifically sound. However, the presentation of information by this paper must be improved before it can be published. My specific comments below:

Figure 1: It is unclear to me why cancer mortality was categorized into two groups : lung cancer and other cancer - is lung cancer the largest cause of cancer deaths among all kinds?

Figure 2: All text is not very visible in this figure. I also struggled a lot with understanding the meaning of the numbers. For instance, the note of the figure states that "the exact percentages are given in the bar under the respective map picture", but in the first chart titled "age (years"), looks like the number in the bar are ages, not percentages. If I read that correctly, what do the red parts in the age map mean? Also, what was the denominator used to calculate all the percentages? All death records or records in each cluster? The same question applies to Figures 3 and 4.

Line 146: "had a significantly higher mean age" - what was that mean age?

Line 147: what does local residents mean here? In another word, how does a death record in the United States define a person as a local resident or something else (a foreigner maybe?)?

Line 153: Why cardiac arrest is considered as a type of cancer?

Line 156: "significantly lower mean age" - what was that mean age?

Line 173: Why cardiac arrest is considered as a "respiratory disease"?

Sections 3.2.1 to 3.2.16: All these sections (long text) use the same format to report the same type of statistics. I think these 16 sections can be replaced by a large table, which will make it a lot easier for the readers to follow and find the key information they are looking for.

Other minor comments:

Line 60: why "big data" was capitalized?

Table A3: several formatting issues

Author Response

C11. Figure 1: It is unclear to me why cancer mortality was categorized into two groups : lung cancer and other cancer - is lung cancer the largest cause of cancer deaths among all kinds?

R11. Yes, 1 725 368 people had lung cancer and other cancers were less prevalent (see column B in table A4.

C12. Figure 2: All text is not very visible in this figure. I also struggled a lot with understanding the meaning of the numbers. For instance, the note of the figure states that "the exact percentages are given in the bar under the respective map picture", but in the first chart titled "age (years"), looks like the number in the bar are ages, not percentages. If I read that correctly, what do the red parts in the age map mean? Also, what was the denominator used to calculate all the percentages? All death records or records in each cluster? The same question applies to Figures 3 and 4.

R12. Thank you very much for pointing out the not fitting description for the age variable. We adapted the legend of figure 2:

“Age: Red parts of the map consist of persons with a high average age at death, whereas blue parts consist of persons with a low average age at death. The exact average ages are given in the bar under the map picture. Gender, race, education, resident status, marital status: Red parts of the map consist of high percentages of persons of the indicated gender, race, education, resident status and marital status, whereas blue parts consist of low percentages of the indicated gender, race, education, resident status and marital status. The exact percentages are given in the bar under the respective map picture.”

We have added to the legends of figure 3 and 4: “The exact percentages are given in the bar under the respective map picture.”.

If needed, we can split figure 2 into two figures to make the text larger.

The denominator is the number of records in the corresponding SOM-unit (node).  Each node stands for several very similar persons. The depicted values are the average values of all persons in the node (e.g. average age, percentage of females, etc.). To make that more clear we have added to 2.3 Statistics: 

“A SOM consists of units on a grid (nodes), where each node represents a group of very similar data records. Thereby, the SOM generates a nonlinear representation of the data distribution, which can be used to identify homogenous data groups and perform statistical analyses on them[16].

[…]

Attributes are visualized by depicting the average value for each node on a fitting color scale.” (lines 98-105 of the revised manuscript)

C13. Line 146: "had a significantly higher mean age" - what was that mean age?

R13. The mean age in this cluster was 75.7 (Table A4 cell Y5) while the mean age for the whole group was 73.1 years (Table A4 cell B5). While alle data are shown in Table A4 we refer to this table instead of repeating all numbers in the result section.

C14. Line 147: what does local residents mean here? In another word, how does a death record in the United States define a person as a local resident or something else (a foreigner maybe?)?

R14. We have explained the term local resident: “For resident status, the two categories 2 (intrastate residents) and 3 (interstate residents) were combined into non-local US resident,while other people were categorized as local residents (resident of the state and county in which he or she died) or foreign residents.” (line 591-594 of the revised manuscript)

C15. Line 153: Why cardiac arrest is considered as a type of cancer?

R15. Cardiac arrest is not considered as a type of cancer, but it’s the fifth most prevalent health condition associated with death in the cluster ‘other cancer’. The first four most prevalent health conditions were different types of cancer. “The top-5 most prevalent health conditions associated with death in this cluster were: 1) other cancer (15.8%); 2) breast cancer (13.6%); 3) prostate cancer (10.9%); 4) lung cancer (9.9%); and 5) cardiac arrest (9.6%).” (line 170-172)

The people in the cluster ‘other cancer’ all had some sort of cancer, but for some of them also other health issues were reported in addition to cancer, like infections or ‘cardiac arrest’. Please also see 4.3 Methodological considerations concerning ‘cardiac arrest’.

C16. Line 156: "significantly lower mean age" - what was that mean age?

R16. Mean age for this cluster was 71.8 years (Table A4, cell W5)

C17. Line 173: Why cardiac arrest is considered as a "respiratory disease"?

R17. Cardiac arrest is not considered as a respiratory disease. Please see R15.

C18. Sections 3.2.1 to 3.2.16: All these sections (long text) use the same format to report the same type of statistics. I think these 16 sections can be replaced by a large table, which will make it a lot easier for the readers to follow and find the key information they are looking for.

R18. The reviewer is correct. All these data are available in Table A4. To make it easier for readers to understand Table A4, we have described the first 5 clusters in the main manuscript and we have described the other 11 clusters in Appendix B.

C19. Line 60: why "big data" was capitalized?

R19. We have changed “Big Data” in “big data”. (line 60 of the revised manuscript)

C20. Table A3: several formatting issues

R20. Table A3 was adjusted.

Reviewer 3 Report

I read the paper with great interest and found it to be a nice tour de force on big data analysis. Just the fact that 27.5M records can be analysed at once in an effective manner deserves appreciation. Although the paper is fine in it current form, I have a few remarks regarding the focus and overall presentation of the study.

The paper tries to balance both the methodology and content but faces some difficulties as it does not specifically choosing either of them. If the focus is on the method and its application in epidemiology, a more thorough and technical description of method and software should be given. This would also result in a more constrained focus in the example analysis (e.g. analysis on socio-demographic variations in multimorbidity associated with death). Even if the authors' focus is on epidemiology, the current scope is a bit too wide in my opinion. I believe that the place and manner of death along with underlying causes could very well be presented in separate analysis and current paper would do fine with "just" the socio-demographic patterns of the clusters. 

This would leave more space in the main text for important details  (Figures A1 & A2) and allow for separate analysis of data for men and women. The latter is in my opinion a potential shortcoming of current study as both morbidity and mortality differs by sex. At least authors decision not to so it should be discussed in the paper and data on men should be added to table 1 and Figure 2.

The figures use varying scales for different variables and even for different categories. Although the ranges may vary, the automatic color-coding may lead to misleading results. Also the figure headings could be more detailed.

The socio-demographic differences between clusters should be covered more thoroughly in the discussion section. There is a subsection on marital status but what about sex, education and ethnicity?

The methodological considerations should also cover, in addition to data-aspects, the role of methodological decisions (e.g. merged analysis of men and women) and aspects of analysis and software.

Author Response

C21. I read the paper with great interest and found it to be a nice tour de force on big data analysis. Just the fact that 27.5M records can be analysed at once in an effective manner deserves appreciation. 

R21. Thank you.

C22. Although the paper is fine in it current form, I have a few remarks regarding the focus and overall presentation of the study.The paper tries to balance both the methodology and content but faces some difficulties as it does not specifically choosing either of them. If the focus is on the method and its application in epidemiology, a more thorough and technical description of method and software should be given. This would also result in a more constrained focus in the example analysis (e.g. analysis on socio-demographic variations in multimorbidity associated with death). Even if the authors' focus is on epidemiology, the current scope is a bit too wide in my opinion. I believe that the place and manner of death along with underlying causes could very well be presented in separate analysis and current paper would do fine with "just" the socio-demographic patterns of the clusters. 

This would leave more space in the main text for important details  (Figures A1 & A2) and allow for separate analysis of data for men and women. The latter is in my opinion a potential shortcoming of current study as both morbidity and mortality differs by sex. At least authors decision not to so it should be discussed in the paper and data on men should be added to table 1 and Figure 2.

R12. Thank you for these comments. We have decided to focus on content instead of the methodology. While place and manner of death are important to consider in developing palliative care programmes, we have decided to show these data in the current manuscript. We agree that sex is an important variable to consider in the analyses. Therefore, sex is considered in all analyses. We have decided not to stratify all analyses by sex, because this would result in three times the current number of tables and figures (for the whole group, for men, and for women). Differences between sexes can be seen in the current analyses. For example, figure 2 shows that men are more likely than women to die from open wound / suffocation or other physical harm, while women are more likely to die from neurodegenerative or other geriatric diseases. For each cluster, gender differences are reported in the result section. We have added to the discussion: “First, gender differences were found in all clusters, except chronic kidney disease. For example, men are more likely than women to die from an open wound / suffocation or other physical harm, while women are more likely to die from neurodegenerative or other geriatric diseases.Gender differences are important to consider to prevent disparities in palliative care uptake. Indeed, a recent study showed significant differences between male and female patients with advanced cancer in preference for palliative care. Women were more likely to prefer palliative care then men.[26]” (line 409-415 of the revised manuscript)

C23. The figures use varying scales for different variables and even for different categories. Although the ranges may vary, the automatic color-coding may lead to misleading results. 

R23. We can understand that these varying scales can be difficult to interpret. We use different ranges for the color coding since, in our opinion, it  is the best way to visualize differences for attributes with different incident rates. We could use the same range for all nominal attributes, but in that case rare diseases would show up as a plain blue field and we wouldn’t see that a rare disease is much more common in a specific cluster on the map than in other clusters. To prevent misunderstanding, in each figure legend the color-coding is explained.

C24. Also the figure headings could be more detailed.

R24. We have revised the title of figure 2 into: “Self-organizing maps showing the socio-demographics (age, gender, race, education, resident status and marital status) of the 16 clusters.”

The title of figure 3 has been revised into: “Self-organizing maps showing the place of death (hospital, home, hospice facility, nursing home, other) of the 16 clusters”.

The title of figure 4 has been revised into: “Self-organizing maps showing the manner of death (homicide, suicide, accident, natural death) of the 16 clusters”.

C25. The socio-demographic differences between clusters should be covered more thoroughly in the discussion section. There is a subsection on marital status but what about sex, education and ethnicity?

R25. We have added a paragraph “other socio-demographic differences between clusters” in the discussion:

Other socio-demographic differences between clusters need to be considered to prevent disparities in palliative care uptake as well as to address specific palliative care needs.

First, gender differences were found in all clusters, except chronic kidney disease. For example, men are more likely than women to die from open wound / suffocation or other physical harm, while women are more likely to die from neurodegenerative or other geriatric diseases. Gender differences are important to consider to prevent disparities in palliative care uptake. Indeed, a recent study showed significant differences between male and female patients with advanced cancer in preference for palliative care. Women were more likely to prefer palliative care then men[26].

Second, differences in educational level were found between all clusters. Figure 2 clearly shows that people dying from open wounds, suffocation or poisoning were the least likely to have followed only primary education. Patients dying from other cancer were the most likely to have followed tertiary education, while patients dying from respiratory diseases were the least likely to have followed tertiary education. Individuals with lower educational attainment are at higher risk for health illiteracy, and may have skills to find, understand and apply information about healthcare. Health illiteracy, therefore, negatively affects the ability to participate in shared-decision making as well as advance care planning. Therefore, clinicians need other strategies and/or tools to support communication and decision-making in palliative care in people with health illiteracy[27].

Third, racial differences should be considered. In fact, Native Americans were overrepresented in the clusters liver disease and other physical harm compared to the other clusters. People who are Asian or Pacific Islander were overrepresented in the clusters stroke and gastro-intestinal cancer compared to other clusters, while black people were overrepresented in the cluster open wound/suffocation and underrepresented in the clusters neurodegenerative disease and other geriatric diseases. Race seems to influence the provision of palliative care as well as healthcare intensity at the end of life, with racial/ethnic minorities receiving higher intensity care, such as ICU admission or life-sustaining treatments at the end of life[28]. Among patients with cancer, being black, Hispanic or Asian is associated with dying in the hospital[29]. A US study showed that black children dying from cancer more often received CPR then white children. Hispanic children less frequently received cancer therapy with 28 days prior to death then non-Hispanic children.[30]Moreover, palliative care clinicians seem more reluctant to discuss prognosis with black or Latino patients with cancer then with white patients[31]. Several explanations for these racial differences have been hypothesized, including differences in preferences regarding life-sustaining treatments, but also a lower likelihood of end-of-life preferences being discussed and honored[30].

Fourth, the present study also showed differences in marital status between clusters, which may need to be considered in organizing palliative care. For example, subjects in clusters ‘Gastrointestinal cancer’ (49%), ‘Lung cancer’ (49%), and ‘Other cancer’ (50%) were more often married at time of death, while subjects in the clusters ‘Geriatric disease’ (58%) and ‘Neurodegenerative disease’ (50%) were more often widowed. A study performed in the United Kingdom showed that older people live years (men: 2.4 years; women 3.0 years) with substantial care needs[32]. Partners of persons with disabilities often contribute significantly in providing care, but burden for family caregivers can be considerable[33]. Persons with dementia are less likely to have an informal caregiver available[34]. On the other hand, family caregivers of persons with dementia and disabilities experienced the highest burden[34]. Palliative care interventions have been shown to be effective in reducing burden for family caregivers[35]. In an English household survey, family caregivers at the end of life were more likely to be willing to care again under the same circumstances if the decedent had had access to palliative care services[36].”. (line407-462 of the revised manuscript)

C26. The methodological considerations should also cover, in addition to data-aspects, the role of methodological decisions (e.g. merged analysis of men and women) and aspects of analysis and software.

R26. We have added the following to the discussion to address this point: “The strength of the current study does not lie in the novelty of any disease specific findings. The novelty of our manuscript lies in our approach and the strong statistic confirmation of many more or less known facts, derived from findings of often smaller or specialized studies. Indeed, we included data of 27,525,663 death records that represents a complete population-based sample of death in the US. The strength of this Big Data approach is seeing the "grand picture". This is also why we chose not to stratify the data with respect to gender, race or any other attribute, since we wanted to keep the focus on the entire population, instead of specific subgroups.

The Viscovery software provided innovative data representation and visualization SOMs, clustering as well as statistical profiling of these death records. The use of SOMs in general and the Viscovery software in particular has several advantages for the analysis of this mortality data set. Since the data is large in quantity as well as complexity we wanted a high-performance method to accurately represent the data distribution, while reducing complexity to a handleable degree, which is one of the strengths of self-organizing maps. In addition, this approach allows for innovative visualization to study interdependencies between diseases and other information. Furthermore the map representation can be used to apply hierarchical clustering methods to this huge data set, which would be impractical with classical methods because of their high running time complexity of O(n2) or worse.

The resulting cluster model is the most complete picture we know about common comorbidity groups in US citizens deceased between 2006 and 2016 and their association to socio-demographic patterns and other "meta-data", like place of death. However, the following limitations need to be considered. […]” (line 486-512 of the revised manuscript)